# Accelerating Unbiased LLM Evaluation via Synthetic Feedback

**Zhaoyi Zhou** [1]   **Yuda Song** [1]   **Andrea Zanette** [1]

## Abstract

When developing new large language models (LLMs), a key step is evaluating their final performance, often by computing the win-rate against a reference model based on external feedback. Human feedback is the gold standard, particularly for capturing nuanced qualities like coherence, readability, and alignment with human expectations. However, human evaluations are costly — even for large tech companies — and when conducted with active users, they may negatively impact user experience. A promising alternative is synthetic feedback, where evaluations are conducted by other large language models, including reward models. While this eliminates the need for costly human annotations, it introduces biases that may distort the evaluation process. In this work, we propose a statistically principled framework that integrates human and synthetic feedback to reduce reliance on human annotations while maintaining unbiased win-rate calculations. Our experiments demonstrate a reduction in human annotations by up to 12.2% with an off-the-shelf synthetic evaluator and up to 24.8% with a finetuned variant. Apart from being generalizable, scalable, and free of hyper-parameter tuning, our method offers predictable annotation savings, which can be estimated based on data-dependent characteristics.

## 1. Introduction

Accurately evaluating the performance of large language models (LLMs) is crucial before large-scale deployment. Human judgment remains the gold standard for this evaluation, as it captures nuanced qualities such as coherence, harmlessness, and readability (Bai et al., 2022), while also ensuring alignment with human values (Ouyang et al., 2022). A widely accepted performance metric is the *win rate*, assessed by humans against a reference model (Chiang et al.,

2024). However, this approach demands substantial time and financial resources due to human involvement. When conducted with active system users, it may also diminish user experience, see Figure 2.

In order to mitigate these challenges, recent works have explored cost-efficient alternatives, most notably the use of synthetic feedback generated by other LLMs, a concept often referred to as "LLM-as-a-judge" (Zheng et al., 2023; Dubois et al., 2024), to compute the head-to-head win rate. This approach leverages the computational efficiency of LLMs to evaluate other models, reducing the need for extensive human involvement. Despite its promise, synthetic feedback often introduces biases since LLM can not perfectly reflect human preference, undermining the evaluation reliability (Zheng et al., 2024). As a result, a critical need remains for evaluation methods that reduce the cost of human annotation while maintaining the reliability and generalizability.

Besides replacing the evaluator, recently there has been a growing interest in accelerating LLM evaluation (Ye et al., 2023; Polo et al., 2024a; Zhou et al., 2024) with smaller datasets. However, previous methods only focused on reducing the number of prompts in a specific benchmark with predefined answers (e.g., math problems). Thus it is unclear if these methods generalize or apply to other tasks. For example, in math benchmark it is easy to find some problems that are "representative" of the whole benchmark, but in general the prompts are more diverse and less structured, and sometimes they are generated on the fly, such as when a user interacts with a language model via APIs.

Towards reliable and cost-efficient LLM evaluation, in this work we propose to leverage LLM generated synthetic feedback to reduce the number of human annotations, in the standard head-to-head win rate setting (Chiang et al., 2024). Specifically, we propose *Control Variates Evaluation* (Figure 1 left), an unbiased LLM evaluation method based on the classical control variates technique (Lavenberg & Welch, 1981) that combines human annotations and synthetic feedback. Note that there are previous works (Chaganty et al., 2018; Boyeau et al., 2024) that apply control variates to machine learning evaluation, but they study settings like single-response natural language evaluation or BT modelling (Bradley & Terry, 1952). Therefore, the performance of control variates in head-to-head win rate estimation still

---

[1]Carnegie Mellon University. Correspondence to: Zhaoyi Zhou <zhaoyiz@andrew.cmu.edu>.

*Proceedings of the 42$^{nd}$ International Conference on Machine Learning*, Vancouver, Canada. PMLR 267, 2025. Copyright 2025 by the author(s).

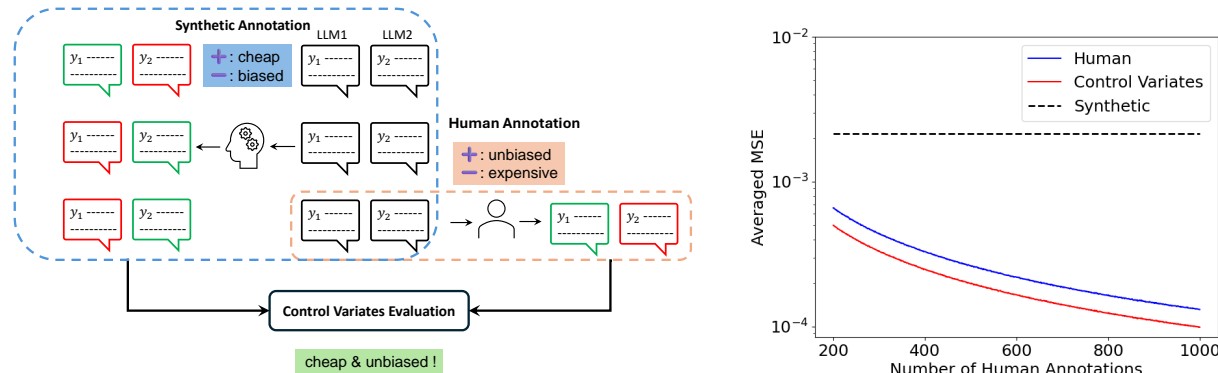

Figure 1: (Left) Illustration of Control Variates Evaluation, which makes use of a possibly inaccurate synthetic evaluator to reduce the variance of evaluation, reducing the need of human annotations while preserving unbiasedness. (Right) Averaged mean square error v.s. number of human annotations for Human Evaluation, Synthetic Evaluation and Control Variates Evaluation using the finetuned Skywork-8B evaluator on Chatbot Arena. The Synthetic Evaluation has high bias, while the bias of Human and Control Variates Evaluations are negligible. Control Variates Evaluation reduces the variance of Human Evaluation.

requires thorough investigation.

In our work, we theoretically show that Control Variates Evaluation enjoys a lower variance, and thus it requires fewer human annotations to achieve the same level of accuracy on the win rate estimation. Empirically, Control Variates Evaluation enjoys significant human annotation saving for various types of synthetic evaluators, from a small reward model with 2B parameters to LLMs such as GPT-4. In addition, we can further reduce human annotations by finetuning the synthetic evaluators on existing human annotations for other LLMs. Note that the cost of control variates is minimal as it only requires some additional synthetic feedbacks, which can be generated at a low cost. Somehow surprisingly, the synthetic evaluators that contribute to such achievement are inaccurate themselves and have high prediction bias (c.f. Figure 1 right).

Besides the advantage of reducing the number of human annotations, Control Variates Evaluation also has a predictable saving, one that can be estimated from the data and one which depends on how strongly the synthetic feedback correlates with human judgments. This is in contrast to the all existing methods that do not provide predictions on the potential saving. Based on the theoretical guarantee, we propose *human annotation saving ratio* as a metric to evaluate our method, which can be computed through a few human annotations without actually running the evaluation. We demonstrate through experiments that this metric perfectly reflects the practical variance reduction effect in Control Variates Evaluation.

In summary, our contribution is three folds:

1. We introduce Control Variates Evaluation to reduce the number of human annotations in head-to-head win rate estimation with zero bias, resulting in a reliable, cost-efficient and task-agnostic LLM evaluation method.
2. We demonstrate the viability of improving human annotation saving through fine-tuning.
3. We propose the human annotation saving ratio as the data-dependent metric to predict the saving in human data when using the Control Variates Evaluation.

We believe our work is a first step towards principled efficient LLM evaluation and can be combined with various existing and future works. Our code is available at https://github.com/Zanette-Labs/control_variates_evaluation.

## 2. Related Work

### 2.1. LLM Evaluation: Metric, Benchmark and Systems

The earliest attempt for LLM evaluation includes rule based metrics such as BLEU (Papineni et al., 2002) and ROUGE (Lin, 2004), which only measures the similarity between the model generation and the reference text. Going beyond rule-based metrics, LLM evaluation has been proposed, with earlier works using LLM to compute similarity (Zhang et al., 2020; Yuan et al., 2021). Recently, LLM-as-a-judge has been proposed to evaluate LLMs (Zheng et al., 2023; Dubois et al., 2024; Gu et al., 2024; Li et al., 2024a; Lan et al., 2024), by querying powerful LLMs to generateli2024generation preference of generations between different models, with the hope that the powerful LLMs can serve as a proxy for human evaluation. Towards real human evaluation, very few public

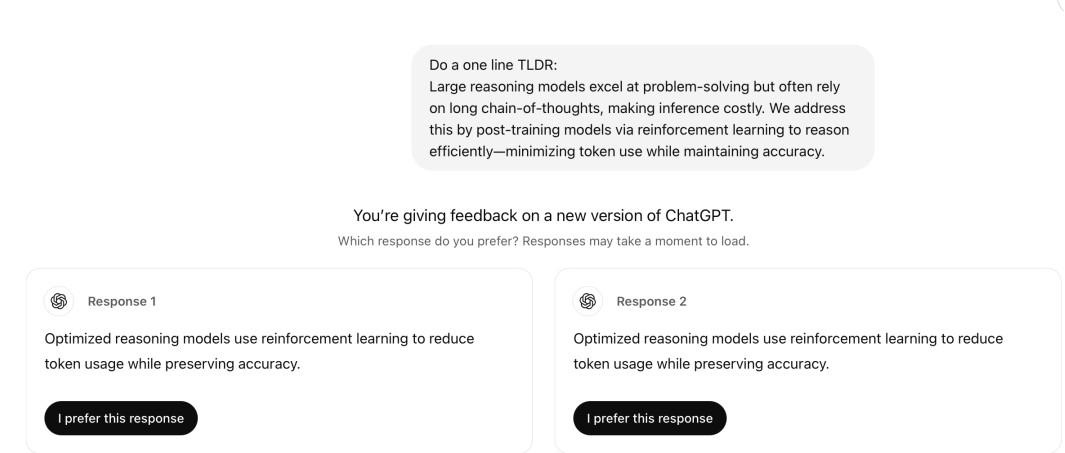

Figure 2: OpenAI's prompting users for feedback; excessive requests may negatively impact user experience.

systems exist due to their high cost and time-consuming nature, with the large-scale community collective effort Chatbot Arena (Chiang et al., 2024) being the most notable one.

## 2.2. Speeding Up LLM Evaluation

Recently there has been a surge of research on speeding up LLM evaluation, with the goal of reducing the cost and time of evaluating LLMs. One approach is to use heuristics to minimize the number of prompts or tasks to evaluate, with the hope that the selected subset can represent the whole distribution of the prompts or tasks (Ye et al., 2023; Perlitz et al., 2023; Polo et al., 2024a).

The other approach is to leverage active learning or bandit algorithms to select a subset of the prompts: (Polo et al., 2024b; Zhou et al., 2024; Li et al., 2024b). However, these methods are still limited by the requirement to operate within a specific benchmark with prefined answers, and thus can not be applied to human evaluation, the focus of our work. In addition to the essential benefit that human evaluation can provide, note that it is more challenging because it is task-agonistic and typically has less structure than any specific benchmark.

## 2.3. Control Variates, Application, and related techniques

Control variates is a well-known variance reduction technique in Monte Carlo sampling (Owen, 2013), with applications to finance (Broadie & Glasserman, 1998; Hesterberg & Nelson, 1998; Kemna & Vorst, 1990; Glasserman, 2004). In recent years, it has also been applied to various areas of machine learning, such as variational inference (Geffner & Domke, 2018; Phan et al., 2023), bandits (Verma

& Hanawal, 2021), optimization (Yuan et al., 2024), computer graphics (Rousselle et al., 2016; Müller et al., 2020). In particular (Chaganty et al., 2018) uses control variates to evaluate natural language metrics, but it is restricted to single response evaluation. In our work, we extend control variates evaluation to pairwise LLM comparison.

Prediction-Powered Inference (PPI, and PPI++) (Angelopoulos et al., 2023a;b; Boyeau et al., 2024) is a related technique which uses variance reduction to improve the MLE objective. (Boyeau et al., 2024) applies PPI++ to estimate practical metrics in machine learning, such as accuracy, correlation and BT model (Bradley & Terry, 1952) in pairwise model comparisons. It differs from our work which conducts an in-depth study of control-variates to accelerate head-to-head win rate estimation.

## 3. Preliminaries

### 3.1. LLM Evaluation

We consider the problem of evaluating LLMs performance through head-to-head comparisons, via human preference judgments. Given a set of prompt $\mathcal{X}$, we compare two LLMs $\ell^1$ and $\ell^2$ by estimating the win rate of $\ell^1$ over $\ell^2$ on $\mathcal{X}$.

Formally, we independently sample a prompt $x \in \mathcal{X}$, and sample two responses $y^1 \sim \ell^1(\cdot \mid x)$ and $y^2 \sim \ell^2(\cdot \mid x)$ from $\ell^1$ and $\ell^2$ respectively. We then ask human annotators to choose the better response with label $z(y^1 \succ y^2)$, where

$$z(y^1 \succ y^2) = \begin{cases} 1 & \text{if } y^1 \text{ is preferred over } y^2, \\ 0 & \text{if } y^2 \text{ is preferred over } y^1, \\ 0.5 & \text{if tie.} \end{cases}$$

We will use the shorthand $z$ sometimes in the rest of the text when the context is clear. The win rate of $\ell^1$ over $\ell^2$ on the

prompt $x$ is defined as

$$p\big(\ell^1 \succ \ell^2\big) := \mathbb{E}_{x,y^1,y^2}\big[z\big(y^1 \succ y^2\big)\big],$$

i.e., the averaged human preference over the prompt set, and $\mathbb{E}_{x,y^1,y^2}[\cdot] := \mathbb{E}_{x\sim\mathrm{Uniform}(\mathcal{X})}\big[\mathbb{E}_{y^1\sim\ell^1(\cdot|x),y^2\sim\ell^2(\cdot|x)}[\cdot]\big]$. To estimate $p(\ell^1 \succ \ell^2)$ empirically, we collect an evaluation dataset $\mathcal{D}^{\mathsf{eval}} = \{(x_i, y_i^1, y_i^2)\}_{i=1}^n$, estimate human preference $z_i = z(y_i^1 \succ y_i^2)$ with $z_i^{\mathsf{em}}$ and output the empirical average $\widehat{p}^{\mathsf{em}}(\ell^1 \succ \ell^2) = \frac{1}{n}\sum_{i=1}^n z_i^{\mathsf{em}}$ as the estimate of the win rate. Our goal is to minimize the number of human annotations involved in the process while keeping $\widehat{p}^{\mathsf{em}}$ close to $p$.

### 3.2. Human and Synthetic Evaluation

*Human Evaluation* annotates every sample $(x_i, y_i^1, y_i^2)$ in $\mathcal{D}^{\mathsf{eval}}$ with human, i.e. let $z_i^{\mathsf{em}} := z_i$. This makes the evaluation unbiased. However, leveraging human annotator is extremely expensive, but without enough amount of samples $n$, the empirical mean $\widehat{p}^{\mathsf{em}}(\ell^1 \succ \ell^2)$ can be very noisy due to high variance from a small sample size.

On the other hand, *Synthetic Evaluation* generates preference estimates $\hat{z}(y_i^1 \succ y_i^2)$ using a reward model or LLM (e.g., GPT-4) (Zheng et al., 2023) on every sample. Although it completely obviates the need for human annotations, the evaluation is biased and can lead to inaccurate win rate prediction.

### 3.3. Other Notations

For two one-dimensional random variables $x$ and $y$, we use $\mathrm{Cov}[x,y]$, $\mathrm{Corr}[x,y]$ to denote the covariance and correlation coefficient between $x$ and $y$, respectively. We use $\mathrm{Var}[x]$ to denote the variance of $x$. Let $\{x_i\}_{i=1}^n$, $\{y_i\}_{i=1}^n$ be samples of $x$ and $y$, respectively, we abuse the notation and use $\mathrm{Var}\big[\{x_i\}_{i=1}^n\big]$ for the empirical variance of $\{x_i\}_{i=1}^n$, and $\mathrm{Cov}\big[\{x_i\}_{i=1}^n, \{y_i\}_{i=1}^n\big]$, $\mathrm{Corr}\big[\{x_i\}_{i=1}^n, \{y_i\}_{i=1}^n\big]$ for the empirical covariance and correlation coefficient between $\{x_i\}_{i=1}^n$ and $\{y_i\}_{i=1}^n$ respectively.

## 4. Efficient LLM Evaluation via Control Variates

In this section, we introduce *Control Variates Evaluation*, which combines human and synthetic annotations to realize a variance-reduced unbiased evaluation method, based on control variates (Lavenberg & Welch, 1981). We first recap the classical control variates method in the context of LLM evaluation, and then formally describe how to adapt the control variates method to make it applicable in practice. Finally, we briefly discuss its application in the LLM-as-a-judge setting (Zheng et al., 2023).

---

**Algorithm 1** Control Variates Evaluation

1: **Input:** Evaluation dataset $\mathcal{D}^{\mathsf{eval}} = \big\{(x_i, y_i^1, y_i^2)\big\}_{i=1}^n$, human annotation budget $k$,
2: **Optional Input:** Finetune dataset $\mathcal{D}^{\mathsf{finetune}} = \big\{(x_j, y_j^1, y_j^2)\big\}_{j=1}^m$ with human annotations $\{z_j\}_{j=1}^m$.
3: (Optional) Finetune the synthetic evaluator on $\mathcal{D}_{\mathsf{finetune}}$.

4: Get synthetic evaluations $\hat{z}_1, \hat{z}_2, \cdots, \hat{z}_n$ on $\mathcal{D}^{\mathsf{eval}}$.
5: Sample $k$ data from $\mathcal{D}^{\mathsf{eval}}$ and get human annotations $z_{i_1}, z_{i_2}, \cdots, z_{i_k}$.
6: Estimate $\mu_{\hat{z}} = \frac{1}{n}\sum_{i=1}^n \hat{z}_i$.
7: Estimate $\alpha$ using $\big\{z_{i_j}\big\}_{j=1}^k$ and $\big\{\hat{z}_{i_j}\big\}_{j=1}^k$ by Equation (2)
8: Output the estimated win rate

$$\frac{1}{k}\sum_{j=1}^k z_{i_j} - \alpha\left(\frac{1}{k}\sum_{j=1}^k \hat{z}_{i_j} - \mu_{\hat{z}}\right).$$

---

### 4.1. Control Variates

Given a sample $(x, y^1, y^2)$ with human preference $z$ and synthetic preference $\hat{z}$, we treat $z$ as the random variable for which we want to estimate its mean. Using $\hat{z}$ as the control variate, the classical control variates approach (Lavenberg & Welch, 1981) constructs a new estimated preference:

$$z^{\mathsf{em}} := z^{\mathsf{cv};\alpha} = z - \alpha(\hat{z} - \mu_{\hat{z}}), \tag{1}$$

where $\mu_{\hat{z}} = \mathbb{E}_{x,y^1,y^2}\big[\hat{z}(y^1 \succ y^2)\big]$ is the **synthetic win rate**, and $\alpha \in \mathbb{R}$ is the **control variates coefficient** used to control the variance of $z^{\mathsf{cv};\alpha}$. Intuitively, $\mu_{\hat{z}}$ cancels out the bias incurred by the control variate $\hat{z}$, keeping the estimate unbiased. In addition, assuming that $\mu_{\hat{z}}$ is known, we can guarantee variance reduction compared to human evaluation, as stated by

**Proposition 4.1** (Control Variates Properties (Lavenberg & Welch, 1981) )**.** *Suppose the expectations, variances, covariances and correlation coefficients, unless otherwise stated, are taken under the distribution $x \sim \mathrm{Uniform}(\mathcal{X})$, $y^1 \sim \ell^1(\cdot \mid x)$, $y^2 \sim \ell^2(\cdot \mid x)$. Then the control variates estimate $z^{\mathsf{cv};\alpha}$ enjoys the following properties*

*(1) (Unbiasedness) For any $\alpha \in \mathbb{R}$, we have*
$\mathbb{E}[z^{\mathsf{cv};\alpha}] = p(\ell^1 \succ \ell^2)$.

*(2) (Variance Reduction) Let $\rho = \mathrm{Corr}[z, \hat{z}]$ be the correlation coefficient between human and synthetic preference. Then we have*

$$\min_{\alpha\in\mathbb{R}} \mathrm{Var}[z^{\mathsf{cv};\alpha}] = \big(1 - \rho^2\big)\mathrm{Var}[z].$$

*The minimum is achieved if and only if $\alpha$ equals*

$$\alpha^* = \frac{\text{Cov}[z, \hat{z}]}{\text{Var}[\hat{z}]}.$$

*(3) (Human Annotation Saving) Given an evaluation dataset $\mathcal{D}_{\text{eval}} = \{(x_i, y_i^1, y_i^2)\}_{i=1}^n$, in which $\{x_i\}_{i=1}^n$ are sampled i.i.d. from $X$, $y_i^1 \sim \ell^1(\cdot \mid x_i)$, $y_i^2 \sim \ell^2(\cdot \mid x_i)$ ($i \in [n]$). Let $\{i_j\}_{j=1}^m$ be independently sampled from $[n]$. Then when $m = (1 - \rho^2)n$, we have*

$$\text{Var}\left[\frac{1}{m}\sum_{j=1}^m z_{i_j}^{\text{cv};\alpha^*}\right] = \text{Var}\left[\frac{1}{n}\sum_{k=1}^n z_k\right]$$

*Here the variance on the right hand side is taken by the randomness of sampling $\{(x_i, y_i^1, y_i^2)\}_{i=1}^n$. The variance on the left hand side is taken by the randomness of sampling $\{(x_i, y_i^1, y_i^2)\}_{i=1}^n$ as well as that of sampling $\{i_j\}_{j=1}^m$.*

We provide the proof in Appendix A for completeness.

**Human annotation saving ratio.** Proposition 4.1 immediately suggests that the control variates method can *reduce the percentage* of human annotations by $\rho^2$ while maintaining the same variance as that of Human Evaluation, with negligible cost of querying the synthetic evaluator. Therefore, $\rho^2$ is an important metrics to measure the variance reduction effect and cost efficiency of control variates method. We formally define it below.

**Definition 4.2** (Human annotation saving ratio). The *human annotation saving ratio* of a synthetic evaluator w.r.t. LLMs $\ell^1, \ell^2$ and prompt set $\mathcal{X}$ is defined as

$$\rho^2 = \left(\text{Corr}_{x,y^1,y^2}[z(y^1 \succ y^2), \hat{z}(y^1 \succ y^2)]\right)^2.$$

Here $z(y^1 \succ y^2)$ is the human preference, and $\hat{z}(y^1 \succ y^2)$ is the synthetic preference. The correlation coefficient is computed under the distribution $x \sim \text{Uniform}(\mathcal{X})$, $y^1 \sim \ell^1(\cdot \mid x)$, $y^2 \sim \ell^2(\cdot \mid x)$.

Nonetheless, to apply control variates approach in the context of LLM evaluation, we still face the following challenges: 1) How to estimate the synthetic win rate $\mu_{\hat{z}}$? 2) How to compute the correlation coefficient $\alpha$ in practice to achieve the lowest variance? 3) How to improve the correlation coefficient if the off-the-shelf automatic evaluator does not give a satisfactory human annotation saving ratio? In the following, we discuss how to construct the control variates for LLM evaluation.

### 4.2. Control Variates Evaluation

Algorithm 1 describes the full procedure of control variates evaluation. Same as other evaluation methods,

control variates evaluation requires an evaluation dataset $\mathcal{D}^{\text{eval}} = \{(x_i, y_i^1, y_i^2)\}_{i=1}^n$. The Control Variates Evaluation consists of the following steps:

**Synthetic annotation gathering (Line 4).** We generate synthetic preferences $\hat{z}_i \in [0, 1]$ from an automatic annotator for all samples in the evaluation dataset. Synthetic preferences can be generated in various ways depending on the type of automatic annotator. For an LLM annotator like GPT-4, we query the model to directly generate the preference in natural language. If the automatic annotator is a reward model, we can query the rewards $r_i^1$ and $r_i^2$ from the two responses $y_i^1$ and $y_i^2$ respectively, and then compute the synthetic preference as the Bradley-Terry score of the two rewards (Bradley & Terry, 1952), i.e.,

$$\hat{z}_i = \frac{1}{1 + \exp(r_i^2 - r_i^1)}.$$

**Human annotation sampling (Line 5).** We query the human annotator and obtain human preference $z \in \{0, 0.5, 1\}$. Instead of annotating all the samples like in Human Evaluation, we only annotate $k$ samples randomly drawn from the evaluation dataset, in which $k$ is the number of human annotations we want to use. Increasing $k$ lowers the variance of the estimation but raises the cost of evaluation.

**Synthetic win rate estimation (Line 6).** Since $\mu_{\hat{z}}$ is unknown in practice, we estimate it by averaging the synthetic evaluator's preferences on the whole evaluation dataset. In other words, $\mu_{\hat{z}} := \frac{1}{n}\sum_{i=1}^n \hat{z}_i$.

**Control variates coefficient computation (Line 7).** Although Proposition 4.1(2) already shows the optimal $\alpha$, the covariance between human and synthetic annotations as well as the variance of synthetic annotations needs to be estimated via sampling. Since human annotations are involved in the computation, we reuse the human annotations $\{z_{i_j}\}_{j=1}^k$:

$$\alpha := \frac{\text{Cov}\left[\{z_{i_j}\}_{j=1}^k, \{\hat{z}_{i_j}\}_{j=1}^k\right]}{\text{Var}\left[\{\hat{z}_{i_j}\}_{j=1}^k\right]}. \tag{2}$$

It is standard practice in control variates to estimate $\alpha$ with Equation (2) (Owen, 2013, Chapter 8.9). Although it introduces some correlation between $\alpha$ and the final estimator, and thus the estimated win rate in Algorithm 1 is technically biased, the incurred bias is usually negligible, and it is standard practice to ignore such bias (Owen, 2013, Chapter 8.9). We also validate this practice through experiments in Section 5.2.

**Win rate estimation (Line 8).** After we obtain estimations of the synthetic win rate $\mu_{\hat{z}}$, and the control variates coefficient $\alpha$, we can apply Equation (1) to get the variance-reduced preference estimates $\left\{ z_{i_j}^{\mathsf{cv};\alpha} \right\}_{j=1}^{k}$ for the samples we collected with human annotations. Then we output the win rate estimate by taking the average over the preference estimates:

$$\hat{p}^{\mathsf{em}}(\ell^1 \succ \ell^2) = \frac{1}{k} \sum_{j=1}^{k} z_{i_j}^{\mathsf{cv};\alpha}$$

$$= \frac{1}{k} \sum_{j=1}^{k} z_{i_j} - \alpha \left( \frac{1}{k} \sum_{j=1}^{k} \hat{z}_{i_j} - \mu_{\hat{z}} \right). \quad (3)$$

**(Optional) Synthetic evaluator finetuning (Line 3).** On many popular LLM evaluation benchmarks such as Chatbot Arena and MT Bench (Zheng et al., 2023), there are abundant off-the-shelf human annotations for pre-generated language model responses. Now suppose we have a new LLM and we want to compare it with the existing ones in the benchmark. Can we make use of these existing human annotations to help reduce the human annotations needed in Control Variates Evaluation?

Recall that the human annotation saving ratio is $\rho^2$, the square of correlation coefficient between human and synthetic annotations. One natural idea is to raise the correlation coefficient by finetuning the synthetic evaluator with existing human annotations, to save future human annotations.

Formally, suppose that we have a finetune dataset $\mathcal{D}^{\mathsf{finetune}} = \left\{ (x_j, y_j^1, y_j^2) \right\}_{j=1}^{m}$ with precollected human annotations $\{z_j\}_{j=1}^{m}$. We discard the ties and assume $z_j \in \{0, 1\}$ for all $1 \leq j \leq m$. In case that the synthetic evaluator is a reward model, we finetune the evaluator on $\mathcal{D}^{\mathsf{finetune}}$ to maximize the Bradley-Terry score (Bradley & Terry, 1952) on the chosen response:

$$\mathrm{BT}\big(r_j^1, r_j^2, z_j\big) = \frac{z_j}{1 + \exp(r_j^2 - r_j^1)} + \frac{1 - z_j}{1 + \exp(r_j^1 - r_j^2)}.$$

After finetuning, we can expect an increase in the correlation coefficient $\rho$ and thus also the human annotation saving ratio when we want to evaluate the win rate between a new LLM pair on the same benchmark. Note that the *dataset used for finetuning the synthetic annotator contains responses generated by LLMs that are different from the LLMs that we wish to evaluate*, i.e., the responses in evaluation dataset are out of distribution w.r.t. the finetune dataset. We explain in Section 5.1 how to guarantee this in our experiments. Despite the out-of-distribution property, we show in the experiment section (c.f. Section 5.4) that the finetuned model still generalizes well in terms of the correlation coefficient to the human annotations.

**Summary.** We offer several remarks:

- Our construction of control variates is *task-agnostic*, i.e, we do not leverage any specific structure or knowledge of the prompt set $\mathcal{X}$.

- The method is *hyperparameter-free* as parameters for control variates like the synthetic win rate $\mu_{\hat{z}}$ and control variates coefficient $\alpha$ are estimated directly from data. (If fine-tuning is used, one still needs to choose fine-tuning hyper-parameters over a validation dataset)

- The performance of Control Variates Evaluation is *predictable*. By sampling a *small* subset of evaluation data, collecting human and synthetic annotations, and computing the human annotation saving ratio, the reduction in human annotations can be accurately estimated without fully performing the evaluation. In the experiment (cf. Section 5.2), we show that the saving ratio of human annotations correctly predicts the observed saving.

## 5. Experiments

To evaluate the performance of control variates in practice, we conduct experiments on real-world datasets to mainly answer the following questions:

1. How does Control Variates Evaluation compare to Human Evaluation and Synthetic Evaluation (c.f. Section 3.2)?
2. How does the finetuning process of the synthetic evaluator affect the human annotation saving?

### 5.1. Setup

**Synthetic evaluators.** Towards a comprehensive analysis, we experiment with synthetic evaluators across various model types and sizes, including GRM-Gemma-2B-sftreg (**GRM-2B**) (Yang et al., 2024), ArmoRM-Llama3-8B (**ArmoRM-8B**) (Wang et al., 2024), Skywork-Reward-Llama-3.1-8B-v0.2 (**Skywork-8B**) (Liu et al., 2024) as well as **GPT-4** (Achiam et al., 2023).

**Finetuning procedure.** The testing of Control Variates with finetuning (Line 3 of Algorithm 1) is done in a cross-validation manner. Suppose there are $K$ LLMs generating responses in the evaluation dataset. Our finetuning procedure trains $K$ reward models, each by leaving out the data for a specific LLM. That is, for each LLM $k$, we finetune the reward model on the head-to-head comparisons over the remaining $K - 1$ LLMs. This finetuned reward model is then evaluated on the head-to-head comparisons involving LLM $k$ against the other $K - 1$ models. When comparing Control Variates Evaluation with finetuning and Synthetic

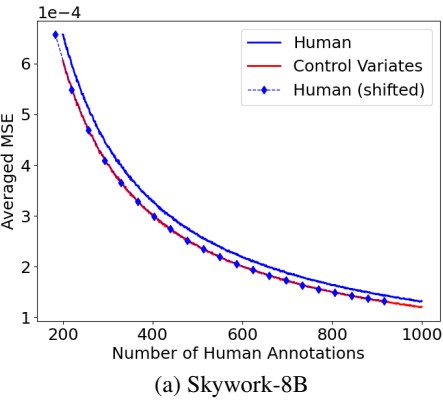 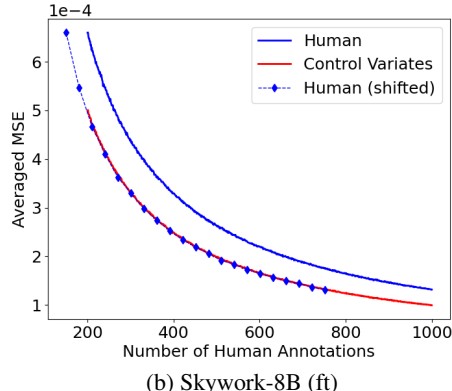

(a) Skywork-8B          (b) Skywork-8B (ft)

Figure 3: Averaged mean-square error versus number of human annotations for Skywork-8B (pretrained and finetuned) on Chatbot Arena. The $x$-coordinate of curves "Human" and "Control Variates" correspond to the number of human annotations (Zheng et al., 2023). The curve "Human (shifted)" is derived by horizontally scaling the Human Evaluation curve by $(1 - s)$, in which $s$ is the averaged human annotation saving ratio in Table 1. The averaged mean-square error of Control Variates Evaluation converges to near 0, indicating that it has negligible bias. The human annotation saving ratio aligns perfectly with the actual variance relationship between Human Evaluation and Control Variates Evaluation.

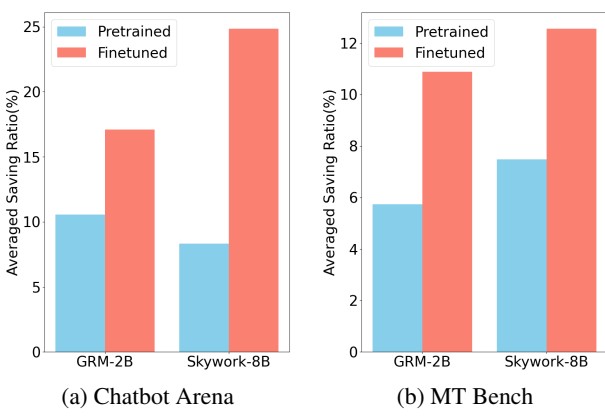

(a) Chatbot Arena      (b) MT Bench

Figure 4: Averaged human annotation saving ratio before and after fine-tuning for GRM-2B and Skywork-8B on Chatbot Arena and MT-Bench. Under all setups, we observe at least 5% increase in the saving ratio.

Evaluation, we apply the same cross-validation procedure to Synthetic Evaluation for a fair comparison.

We tested Control Variates Evaluation with finetuning on GRM-2B and Skywork-8B models, which will be referred to as **GRM-2B (ft)** and **Skywork-8B (ft)** respectively.

**Benchmark.** We choose LLM evaluation datasets with abundant and trustworthy human annotations. The datasets we considered are:

- *ChatBot Arena* (Zheng et al., 2023) contains 33k

human-annotated preferences. The responses are generated by 20 models, i.e., 190 LLM pairs in total. There are 121 pairs that have more than 100 annotations.

- *MT Bench* (Zheng et al., 2023) contains about 3.36k human-annotated preferences. The responses are generated by 6 models, i.e., 15 LLM pairs in total. There are 14 pairs that have more than 100 annotations.

### 5.2. Control Variates Evaluation v.s. Human Evaluation

As suggested in Section 4.1, the human annotation saving ratio is a practical metric to measure the performance of Control Variates Evaluation. Therefore, we will first present the human annotation saving ratio on different evaluators and benchmarks. After that, we demonstrate that this theoretical measure matches perfectly with the actual variance reduction effect.

**Human annotation saving ratio on different benchmarks and synthetic evaluators.** For each synthetic evaluator and benchmark, we test the human annotation saving ratio on every LLM pair that have at least 100 human annotations. In order to clearly present the result, we take the mean of the ratios across different LLM pairs to get the average human annotation saving ratio of that evaluator on the benchmark. The result is presented in Table 1. We defer the human annotation saving ratio on each LLM pair in Appendix C.3.

For off-the-shelf evaluators, GPT-4 achieves high saving ratio on both benchmarks. However, an 8B reward model like ArmoRM-8B has comparable performance. Using the finetuning option of Control Variates Evaluation, Skywork-8B

Table 1: Averaged human annotation saving ratio across different synthetic evaluators on Chatbot Arena and MT Bench. The averaged human annotation saving ratio is the mean of human annotation saving ratios on LLM pairs with at least 100 human annotations.

|  | Chatbot Arena | MT Bench |
|---|---|---|
| GRM-2B | 10.6% | 5.7% |
| GRM-2B (ft) | 17.1% | 10.9% |
| Skywork-8B | 8.3% | 7.5% |
| Skywork-8B (ft) | **24.8**% | **12.6**% |
| ArmoRM-8B | 12.2% | 9.6% |
| GPT-4 | 12.2% | 11.9% |

(ft) surpasses the performance of GPT-4 on both benchmarks. With finetune, a small model (GRM-2B (ft)) can also match or outperform GPT-4 in averaged human annotation saving. This means that we can save from 10% to 20% human annotations using an easy-to-deploy reward model at nearly no cost. In fact, the most expensive synthetic evaluator in our experiment, GPT-4, costs just about 1 cent per annotation. This is negligible comparing to human annotation cost.

**Theory matches practice.** We empirically justify that the theoretical human annotation saving ratio aligns well with the practical variance reduction ratio. Besides, we verify the claim in (Owen, 2013, Chapter 8.9) that Equation (2) leads to negligible bias.

First, we measure the estimated mean square error of Human Evaluation and Control Variates Evaluation w.r.t number of human samples for each fixed LLM pair via bootstrapping. That is, we repeatedly run the evaluation method 1000 times with a fixed number of human annotations, collect the output win rate estimates, and compute the mean-square error, where the ground truth win-rate is the averaged human preference on all data of that LLM pair. For Human and Control Variates Evaluation, we run bootstrapping using different numbers of human annotations on different LLM pairs and plot a curve respectively with labels "Human" and "Control Variates" (c.f. Figure 8), in which the $y$-axis is the averaged mean square error of the evaluation on different LLM pairs, and the $x$-axis is the number of human annotations.

Theoretically, the mean-square error can be decomposed into the square of evaluation bias and the variance. Therefore, the mean-square error curve still effectively reflects the variance reduction tendency as the number of human annotations increases, and when the number approaches infinity, we can extract the bias of the evaluation through the limit of mean square error.

Then, we shift the $x$-axis of the Human Evaluation as follows. Suppose $s$ is the averaged human annotation saving ratio we tested in Table 1, and $(x, y)$ is a point on the curve of Human Evaluation. Then we shift point $(x, y)$ to $(x(1 - s), y)$. After shifting all points of the Human Evaluation, we get a new curve, referred to as *Human (shifted)*. According to Proposition 4.1 (3), the ratio of the number of human annotations in Human Evaluation and Control Variates Evaluation should be $1 : (1 - s)$ so that they have the same variance. So ideally, the shifted curve of Human Evaluation should coincide with the curve of Control Variates Evaluation. We present the bootstrap curves for Skywork-8B with and without the finetuning procedure on Chatbot Arena in Figure 3. The other results are listed in Figure 8 of Appendix.

On all figures, the averaged mean-square error of Control Variates Evaluation converges to near 0, indicating negligible evaluation bias. Furthermore, the shifted curve of Control Variates Evaluation overlaps with that of human evaluation. Therefore, the human annotation saving ratio predicts the actual variance reduction of our algorithm almost perfectly, even if the control variates coefficient $\alpha$ is estimated. This means that we can simply compute the human annotation saving ratio from the correlation coefficient, and then we know whether the synthetic evaluator will bring us the desired variance reduction effect when it is to be used in Control Variates Evaluation.

### 5.3. Control Variates Evaluation v.s. Synthetic Evaluation

In this section, we compare the error in predicting the win rate between the Control Variates Evaluation and Synthetic Evaluation. The error metric is the mean square error with respect to the ground truth win-rate, which we approximate with the averaged human annotations on all samples of each head-to-head comparison. For Control Variates Evaluation, we use the averaged mean-square error from the previous section. For Synthetic Evaluation, we average the synthetic annotations on all samples of a fixed LLM pair as the predicted win rate and then calculate the mean square error. We also include the averaged mean-square error of human for convenience of comparison.

Figure 1 (right) presents the result of finetuned Skywork-8B, and Figure 5 presents that of GPT-4, both on Chatbot Arena. Other results are deferred to Figure 7. Although GPT-4 is claimed to be an accurate evaluator (Zheng et al., 2023), it still has a significantly high error compared to Control Variates and Human Evaluation. Similarly, even if we finetune a reward model like Skywork-8B (ft), it also suffers from high error if used in Synthetic Evaluation alone. However, these evaluators can be incorporated into Control Variates Evaluation to achieve much lower evaluation error.

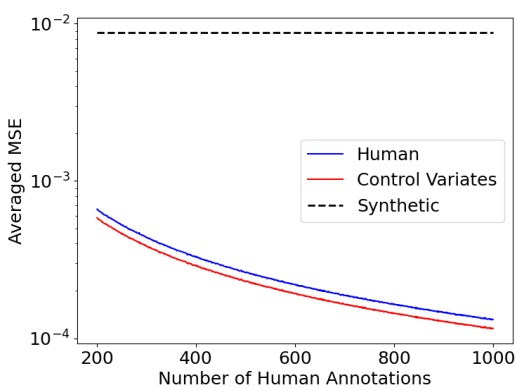

Figure 5: Average mean square error versus number of human annotations for GPT-4 evaluator on Chatbot Arena (Zheng et al., 2023). Note that even GPT-4 has high bias if used alone for Synthetic Evaluation.

### 5.4. How does Finetuning Improve Control Variates Evaluation?

We visualize the averaged human annotation saving ratio before and after finetuning for GRM-2B and Skywork-8B on Chatbot-Arena and MT-Bench in Figure 4. For all experiments, the finetuning procedure provides at least 5% more saving ratio. Specifically, for Skywork-8B on Chatbot Arena, the saving ratio nearly triples.

On the other hand, finetuning indeed introduces additional computation requirement. For instance, the fine-tuning of Skywork-8B model requires four H100 GPUs with 80GB of GRAM. Regarding whether to finetune the evaluator or not, there are two major considerations. The first one is the human annotation saving ratio on the pretrained evaluator. If it is not satisfactory, finetuning can improve the human annotation saving ratio if a finetune dataset is available. The other consideration is the number of future tasks, as this is a trade-off between future savings in human annotation cost and the current additional cost of finetuning computation. If there are many future models to evaluate, then finetuning is beneficial because the savings generalize to unseen models.

### 5.5. Other Applications of Control Variates Evaluation

Control Variates Evaluation can be similarly applied in the LLM-as-a-judge setting. The difference is that the human annotator is replaced with a strong LLM evaluator, and a smaller, cheaper model plays the role of the synthetic evaluator, to save the cost of querying the expensive model.

We set GPT-4 as the strong evaluator and test the averaged human annotation saving ratio in the scenario of LLM-as-a-judge, as shown in Table 2. A 2B reward model like GRM-2B can achieve over 20% saving of GPT-4 annotation

Table 2: Averaged strong evaluator's sample saving in LLM-as-a-judge using control variates evaluation. The strong evaluator is GPT-4.

| Weak Evaluator | Chatbot Arena | MT Bench |
|---|---|---|
| GRM 2B sftreg | 22.8% | 14.6% |
| Skywork 8B | 13.5% | 15.1% |
| ArmoRM 8B | 16.0% | 18.8% |

on Chatbot Arena and nearly 15% saving on MT Bench. This can save the cost in LLM-as-a-judge.

In addition, we provide the experiment result of Control Variates Evaluation for single response evaluation tasks in Appendix C.4.

## 6. Conclusion

In this work, we propose Control Variates Evaluation to reduce human annotation costs while maintaining unbiasedness. Our method demonstrates significant savings in human annotations across benchmarks like Chatbot Arena and MT Bench, aligning well with theoretical predictions. This provides a scalable and cost-effective alternative to full human evaluation without compromising reliability.

We only study the most canonical evaluation of head-to-head win rate between two LLMs, and it is an interesting future direction to explore more nuanced human evaluation metrics and complex evaluation settings, including multi-model ranking and fine-grained assessments. Other future work can focus on improving synthetic feedback through adaptive selection or ensembling multiple evaluators.

## Acknowledgement

This work used DeltaAI at NCSA through allocation CIS240294: Advancing Large Language Model Alignment through Hybrid Feedback Integration from the Advanced Cyberinfrastructure Coordination Ecosystem: Services & Support (ACCESS) program (Boerner et al., 2023), which is supported by U.S. National Science Foundation grants #2138259, #2138286, #2138307, #2137603, and #2138296.

We also thank the ICML 2025 program chair as well as Ojash Neopan for directing us to Prediction-Powered Inference (PPI), an important and relevant line of our work.

## Impact Statement

This work seeks to accelerate LLM evaluation while preserving its unbiasedness. The societal and ethical impact aligns with that of most LLM evaluation research, which has been widely discussed.

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

# A. Proof of Proposition 4.1

Note that all expectations, variances, covariances and correlation coefficients in this section are taken under the distribution $x \sim \text{Uniform}(\mathcal{X})$, $y^1 \sim \ell^1(\cdot \mid x)$, $y^2 \sim \ell^2(\cdot \mid x)$.

**Proof of unbiasedness**   We have

$$
\begin{aligned}
\mathbb{E}_{x,y^1,y^2}\left[z^{\mathsf{cv},\alpha}\right] &= \mathbb{E}_{x,y^1,y^2}\left[z - \alpha\left(\hat{z} - \mu_{\hat{z}}\right)\right] \\
&= \mathbb{E}_{x,y^1,y^2}[z] - \alpha\left(\mathbb{E}_{x,y^1,y^2}[\hat{z}] - \mu_{\hat{z}}\right) \\
&= \mathbb{E}_{x,y^1,y^2}[z] \\
&= p\left(\ell^1 \succ \ell^2\right).
\end{aligned}
$$

**Proof of variance reduction**   We have

$$
\begin{aligned}
\text{Var}_{x,y^1,y^2}\left[z^{\mathsf{cv},\alpha}\right] &= \text{Var}_{x,y^1,y^2}\left[z - \alpha\left(\hat{z} - \mu_{\hat{z}}\right)\right] \\
&= \text{Var}_{x,y^1,y^2}\left[z\right] - 2\alpha\text{Cov}_{x,y^1,y^2}\left[z, \left(\hat{z} - \mu_{\hat{z}}\right)\right] + \alpha^2\text{Var}_{x,y^1,y^2}\left[\hat{z} - \mu_{\hat{z}}\right] \\
&= \text{Var}_{x,y^1,y^2}\left[z\right] - 2\alpha\text{Cov}_{x,y^1,y^2}\left[z, \hat{z}\right] + \alpha^2\text{Var}_{x,y^1,y^2}\left[\hat{z}\right] \\
&= \text{Var}_{x,y^1,y^2}\left[\hat{z}\right]\left(\alpha - \frac{\text{Cov}_{x,y^1,y^2}\left[z, \hat{z}\right]}{\text{Var}_{x,y^1,y^2}\left[\hat{z}\right]}\right)^2 + \text{Var}_{x,y^1,y^2}\left[z\right] - \frac{\left(\text{Cov}_{x,y^1,y^2}\left[z, \hat{z}\right]\right)^2}{\text{Var}_{x,y^1,y^2}\left[\hat{z}\right]} \\
&\geq \text{Var}_{x,y^1,y^2}\left[z\right] - \frac{\left(\text{Cov}_{x,y^1,y^2}\left[z, \hat{z}\right]\right)^2}{\text{Var}_{x,y^1,y^2}\left[\hat{z}\right]}.
\end{aligned}
$$

The equality holds if and only if $\alpha = \frac{\text{Cov}_{x,y^1,y^2}[z,\hat{z}]}{\text{Var}_{x,y^1,y^2}[\hat{z}]}$. To further simplify the formula, recall that

$$
\rho^2 = \left(\text{Corr}_{x,y^1,y^2}\left[z, \hat{z}\right]\right)^2 = \frac{\left(\text{Cov}_{x,y^1,y^2}\left[z, \hat{z}\right]\right)^2}{\text{Var}_{x,y^1,y^2}[z] \cdot \text{Var}_{x,y^1,y^2}[\hat{z}]}.
$$

Therefore we have

$$
\begin{aligned}
\text{Var}_{x,y^1,y^2}\left[z^{\mathsf{cv},\alpha}\right] &\geq \text{Var}_{x,y^1,y^2}\left[z\right] - \frac{\left(\text{Cov}_{x,y^1,y^2}\left[z, \hat{z}\right]\right)^2}{\text{Var}_{x,y^1,y^2}\left[\hat{z}\right]} \\
&= \text{Var}_{x,y^1,y^2}\left[z\right] - \rho^2\text{Var}_{x,y^1,y^2}\left[z\right] \\
&= (1 - \rho^2)\text{Var}_{x,y^1,y^2}\left[z\right].
\end{aligned}
$$

The optimality point is $\alpha^* = \frac{\text{Cov}_{x,y^1,y^2}[z,\hat{z}]}{\text{Var}_{x,y^1,y^2}[\hat{z}]}$.

**Proof of human annotation saving**   Since all samples are i.i.d., we have

$$
\begin{aligned}
\text{Var}\left[\frac{1}{m}\sum_{j=1}^{m} z_{i_j}^{\mathsf{cv};\alpha}\right] &= \frac{1}{m}\text{Var}_{x,y^1,y^2}[z^{\mathsf{cv},\alpha^*}] \\
&= \frac{1}{m}(1 - \rho^2)\text{Var}_{x,y^1,y^2}[z] \\
&= \frac{1}{n}\text{Var}_{x,y^1,y^2}[z] \\
&= \text{Var}\left[\frac{1}{n}\sum_{i=1}^{n} z_i\right].
\end{aligned}
$$

# B. Experiment Details

## B.1. Hyperparameters

The Control Variates Evaluation Algorithm 1 has no hyperparameters except for the optional finetuning procedure. When finetuning Skywork-8B and GRM-2B on Chatbot Arena and MT Bench, we use global batch size 32 and train for 1 epoch. The finetuning of GRM-2B on Chatbot Arena uses learning rate 1e-6, others all use learning rate 3e-6.

To determine the optimal hyperparameters for finetuning, we conduct a systematic search over a range of learning rates and batch sizes. For instance, when we finetune Skywork-8B on Chatbot Arena, we follow these steps:

(1) We sort the LLM models in Chatbot Arena in alphabetical order and select the first model, RMKV-4-Raven-14B, as the holdout model to split train and test dataset.

(2) We tested learning rates in $\{1\times10^{-7}, 3\times10^{-7}, 1\times10^{-6}, 3\times10^{-6}, 1\times10^{-5}, 3\times10^{-5}\}$ and batch sizes in $\{32, 64, 128\}$. For each hyperparameter combination, we finetune for one epoch and record the final test accuracy.

(3) The combination yielding the highest final test accuracy is selected as the optimal hyperparameter setting. We use the chosen hyperparameter setting to finetune Skywork-8B on all other holdout models.

The similar procedure applies when we finetune other synthetic evaluators on other benchmarks.

## B.2. Hardware

The experiments are run on H100 GPUs. Finetuning Skywork-8B requires 4 GPUs. Finetuning GRM-2B as well as the collection of synthetic annotations can all be done on 1 GPU.

## B.3. Prompt Template

We use the GPT-4 annotations for MT-Bench from the Hugging Face repository https://huggingface.co/datasets/lmsys/mt_bench_human_judgments/viewer/default/gpt4_pair.

We follow the prompt template in (Zheng et al., 2023, Figure 5, Appendix A) to get GPT-4 annotations in Chatbot Arena. We provide the template in Figure 6 for completeness.

# C. Additional Experiment Results

## C.1. Bias of Synthetic Evaluation

As described in Section 5.3, we measure the averaged mean square error of Human Evaluation, Synthetic Evaluation and Control Variates Evaluation on different evaluators and datasets, as shown in Figure 7. The Synthetic Evaluation has a significantly high bias, while the error of both Human Evaluation and Control Variates Evaluation converge to zero.

## C.2. Human Annotation Saving Ratio Matches Variance Reduction in Practice

As described in Section 5.2, we measure the averaged mean square error versus number of samples for different evaluators on different datasets. The $x$-coordinate of curves "Human" and "Control Variates" correspond to the number of human annotations (Zheng et al., 2023). The curve "Control Variates (shifted)" is derived by horizontally scaling the Control Variates curve by $1/(1-s)$, in which $s$ is the averaged human annotation saving ratio in Table 1. The human annotation saving ratio aligns perfectly with the actual variance relationship between Human Evaluation and Control Variates Evaluation.

## C.3. Human Annotation Saving Ratio on Each LLM pair

We visualize the human annotation ratio (in percentage) on each LLM pair that we use to compute the averaged human annotation saving ratio in Table 1. The results are shown in Figures 9 and 10. For a pretrained evaluator, each entry of the matrix is the human annotation saving ratio (in percentage) on that LLM pair. For a finetuned evaluator, each entry of the matrix is the human annotation saving ratio (in percentage) on the corresponding LLM pair, in which the LLM on the row is the left-out LLM, while the LLM on the column is used in finetuning. Please refer to Section 5.1 for the details

---

**Prompt for GPT-4 Annotations in Chatbot Arena**

```
[System]
Please act as an impartial judge and evaluate the quality of the responses provided
by two AI assistants to the user question displayed below. You should choose the
assistant that follows the user's instructions and answers the user's question
better. Your evaluation should consider factors such as the helpfulness, relevance,
accuracy, depth, creativity, and level of detail of their responses. Begin your
valuation by comparing the two responses and provide a short explanation. Avoid any
position biases and ensure that the order in which the responses were presented does
 not influence your decision. Do not allow the length of the responses to influence
your evaluation. Do not favor certain names of the assistants. Be as objective as
possible. After providing your explanation, output your final verdict by strictly
following this format: "[[A]]" if assistant A is better, "[[B]]" if assistant B is
better, and "[[C]]" for a tie.

[User Question]
{question}

[The Start of Assistant A's Answer]
{answer_a}
[The End of Assistant A's Answer]

[The Start of Assistant B's Answer]
{answer_b}
[The End of Assistant B's Answer]
```

---

Figure 6: Prompt template from (Zheng et al., 2023, Figure 5, Appendix A), which is used to get GPT-4 annotations in Chatbot Arena.

of finetuning procedure. Therefore, the matrices for pretrained evaluators are symmetric, while they are asymmetric for finetuned evaluators. The diagonal entries are white and do not have values becuase measuring human annotation saving ratio on identical LLMs is meaningless.

Note that there are some additional white entries with no values when testing GPT-4 as the synthetic evaluator. This is because GPT-4 cannot always follow the prompt template, so that sometimes we cannot extract a valid preference out of the output. In case that there are too few samples in an LLM pair, it is likely that we cannot compute a valid human annotation saving ratio.

### C.4. Single Response Evaluation

Besides head-to-head comparison, Control Variates Evaluation also applies smoothly to other evaluation scenarios, e.g., single response evaluation, where a human gives scores to a single LLM generation, instead of giving preference to two LLM generations.

We utilize the validation split of the HelpSteer2 dataset as our benchmark. This split consists of 1.04K samples, each containing a prompt, a response, and five human-annotated attributes: helpfulness, correctness, coherence, complexity, and verbosity. Each attribute is scored from 0 to 4, with higher scores indicating better performance. Our focus is on the helpfulness attribute, as it is the primary metric that reward models are typically trained to evaluate. We employ the Control Variates Evaluation method to predict the average helpfulness score. The human annotation saving ratio is shown in the Table 3.

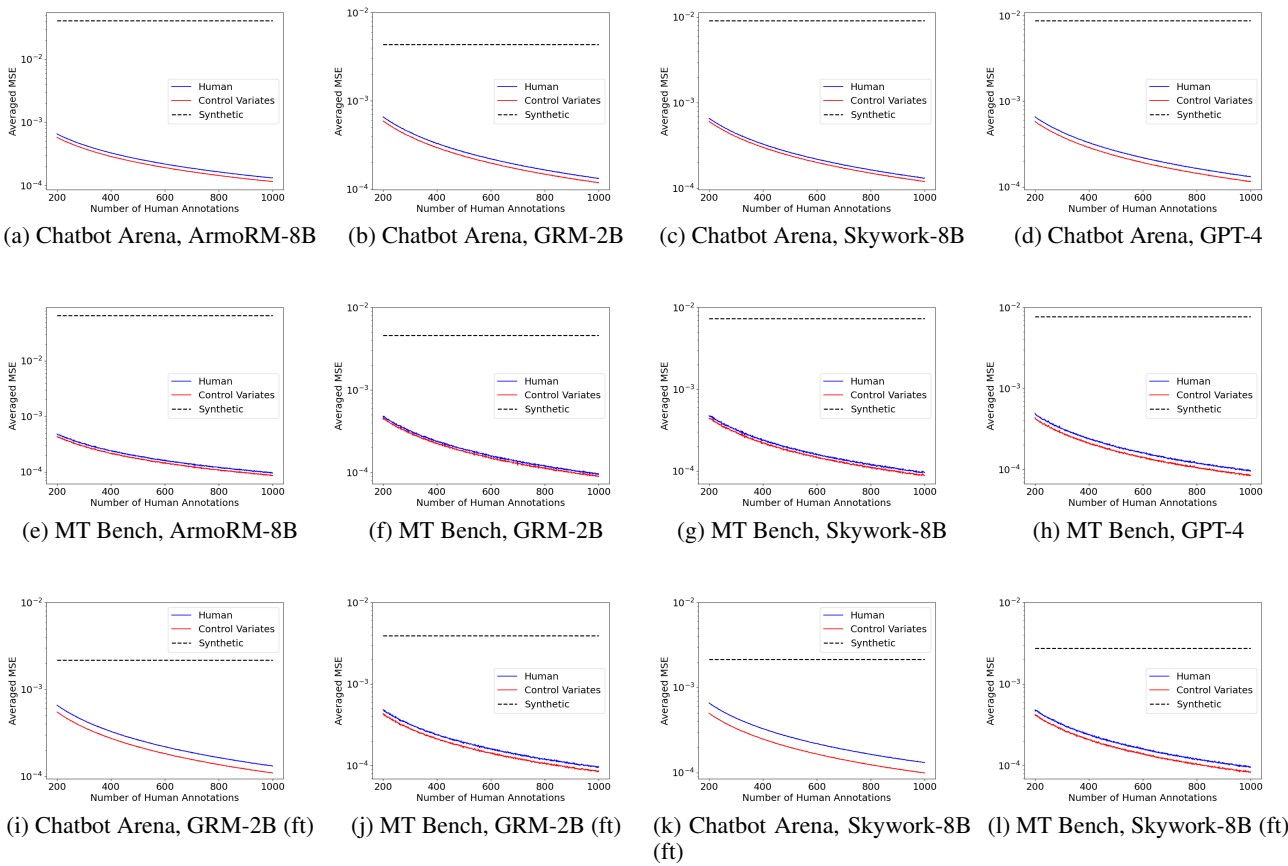

Figure 7: Averaged mean square error of Human Evaluation, Synthetic Evaluation and Control Variates Evaluation on different evaluators and datasets. The Synthetic Evaluation has a significantly high bias, while the error of both Human Evaluation and Control Variates Evaluation converge to zero.

Table 3: Averaged human annotation saving ratio across different synthetic evaluators on Helpsteer2.

|  | Human Annotation Saving Ratio |
| --- | --- |
| GRM-2B | 10.3% |
| Skywork-8B | 21.0% |
| ArmoRM-8B | 14.1% |
| GPT-4o | 27.4% |

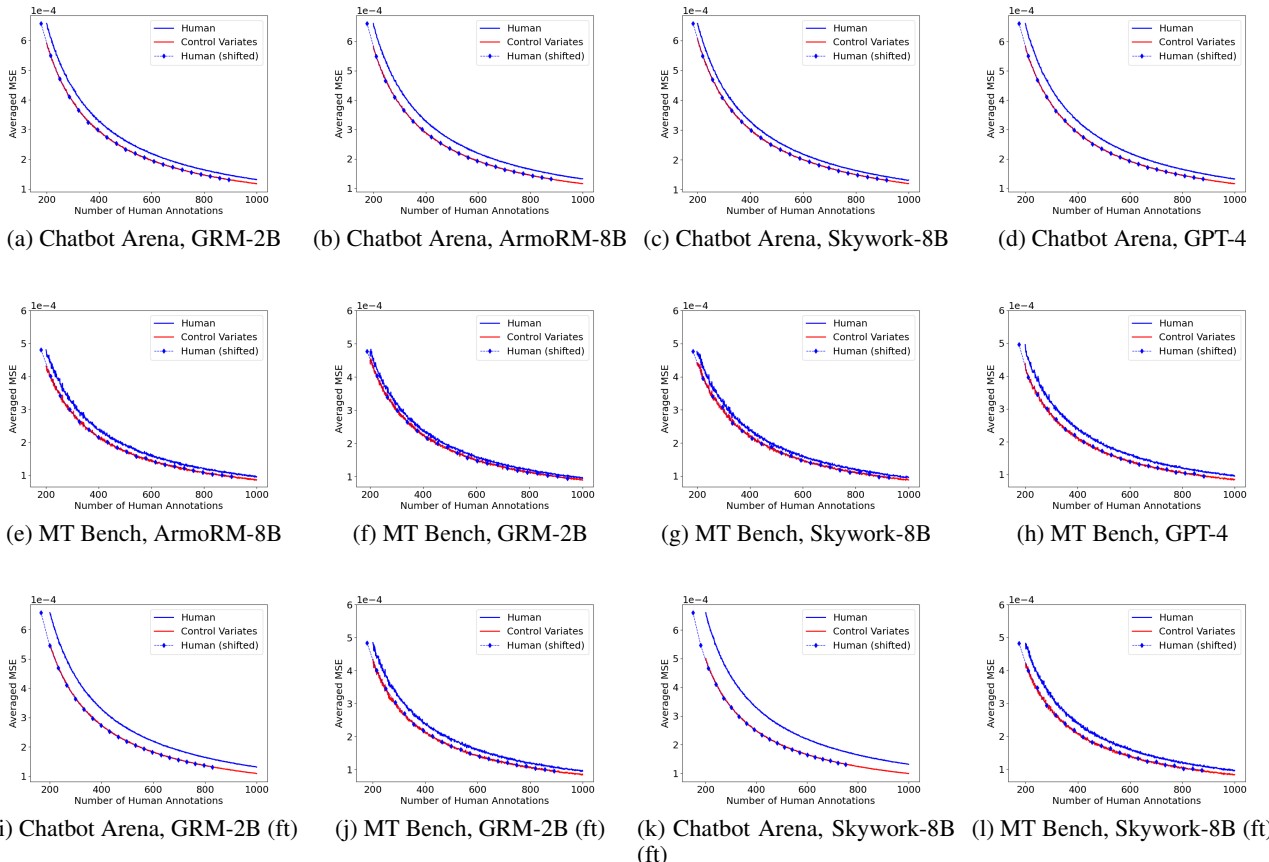

(a) Chatbot Arena, GRM-2B  (b) Chatbot Arena, ArmoRM-8B  (c) Chatbot Arena, Skywork-8B  (d) Chatbot Arena, GPT-4

(e) MT Bench, ArmoRM-8B  (f) MT Bench, GRM-2B  (g) MT Bench, Skywork-8B  (h) MT Bench, GPT-4

(i) Chatbot Arena, GRM-2B (ft)  (j) MT Bench, GRM-2B (ft)  (k) Chatbot Arena, Skywork-8B (ft)  (l) MT Bench, Skywork-8B (ft)

Figure 8: Averaged mean square error versus number of samples for different evaluators on different datasets. The $x$-coordinate of curves "Human" and "Control Variates" correspond to the number of human annotations (Zheng et al., 2023). The curve "Control Variates (shifted)" is derived by horizontally scaling the Control Variates curve by $1/(1 - s)$, in which $s$ is the averaged human annotation saving ratio in Table 1. The human annotation saving ratio aligns perfectly with the actual variance relationship between Human Evaluation and Control Variates Evaluation.

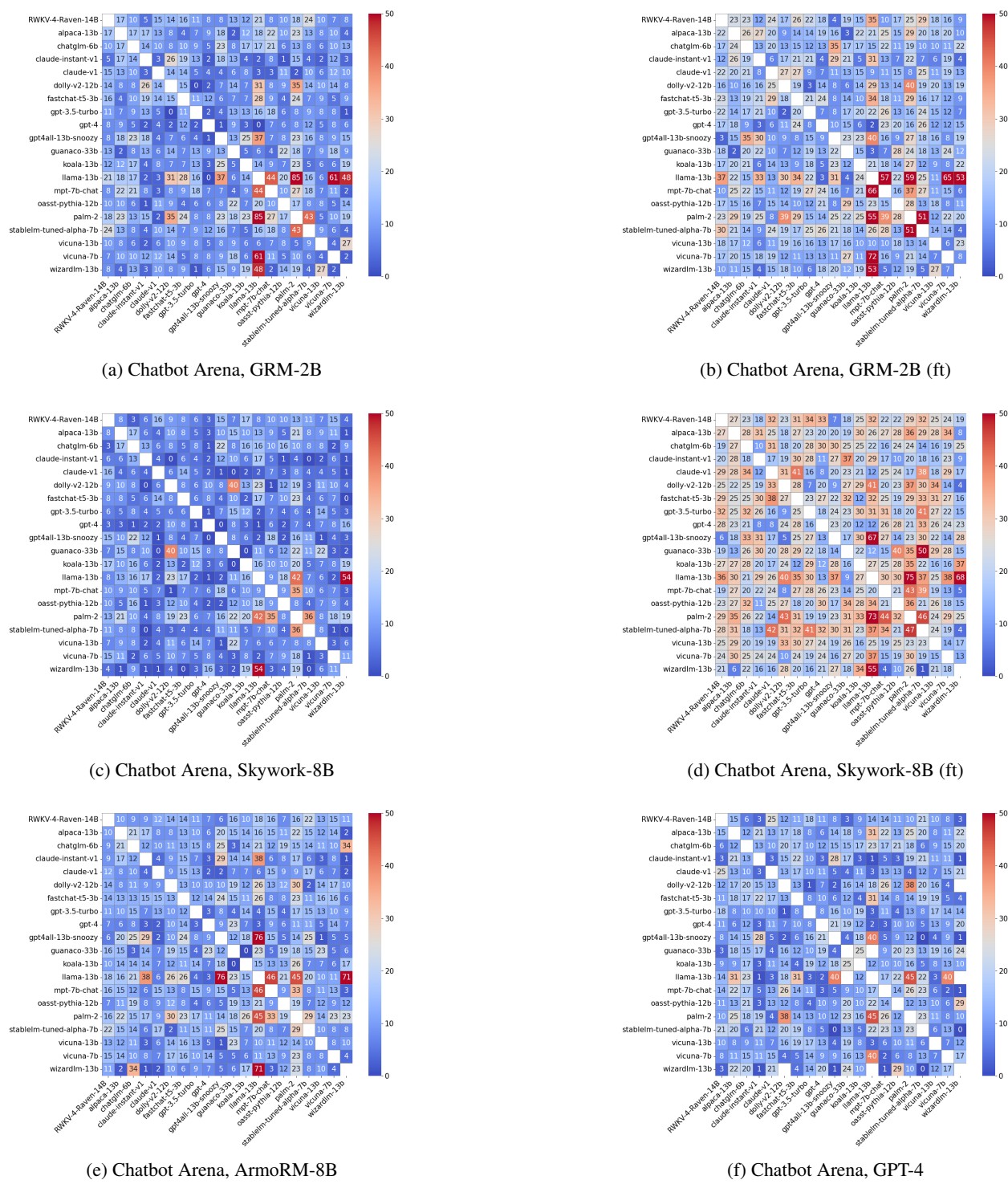

Figure 9: Human annotation saving ratio (in percentage) on each LLM pair for different evaluators on Chatbot Arena. Diagonal entries are white and do not have values because it is meaningless to compute the human annotation saving ratio on two identical LLMs. Non-diagonal white entries in (f) imply an invalid result, because sometimes valid preference cannot be extracted from GPT-4's response.

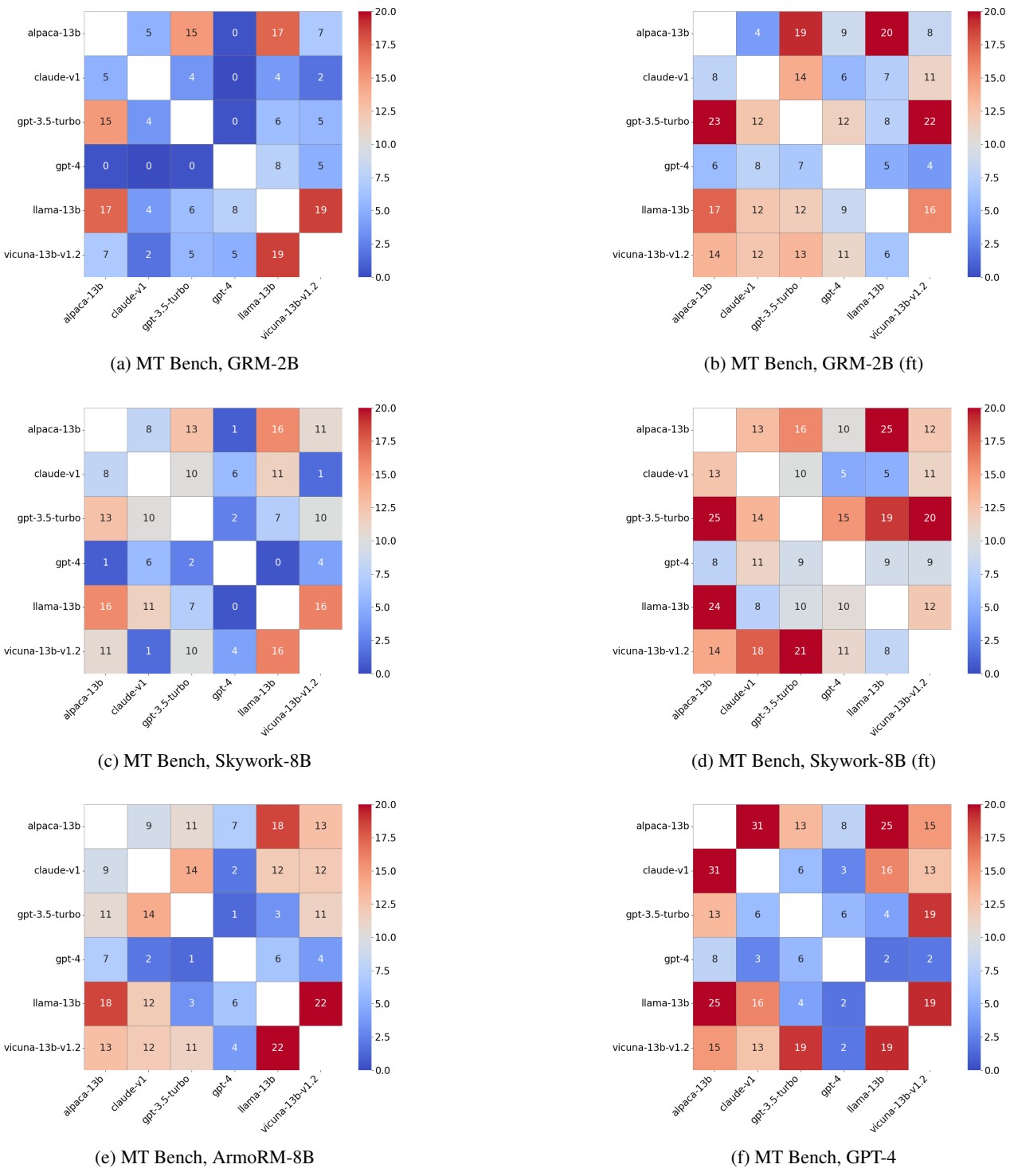

Figure 10: Human annotation saving ratio (in percentage) on each LLM pair for different evaluators on MT Bench. Diagonal entries are white and do not have values because it is meaningless to compute the human annotation saving ratio on two identical LLMs.

