# OpenReview forum: "Accelerating Unbiased LLM Evaluation via Synthetic Feedback"
_ICML.cc/2025/Conference — ICML 2025 poster_

### Official Review · Reviewer_j7BE · 2025-03-03

**Overall Recommendation:** 4

**Summary:**

This paper introduces Control Variates Evaluation, a novel method for unbiased and cost-efficient evaluation of large language models (LLMs) in head-to-head comparisons. The approach leverages synthetic feedback from LLMs, combined with human annotations, to reduce annotation costs while maintaining evaluation reliability. The authors demonstrate that this method reduces the number of required human annotations by up to 24.8% when synthetic feedback is fine-tuned. Theoretical guarantees for variance reduction are provided, and the method is empirically validated against benchmarks such as Chatbot Arena and MT Bench. Additionally, the paper introduces a human annotation saving ratio metric to predict the potential savings.

**Claims And Evidence:**

The claims in the paper are mostly supported by convincing evidence. The authors provide both theoretical analysis and extensive experimental results to validate the effectiveness of the proposed method. Key claims, such as the reduction in human annotations and the alignment between theoretical variance predictions and empirical results, are well-supported. However, some claims about the scalability and generalizability of the method to more complex evaluation tasks (e.g., beyond head-to-head comparisons) are less substantiated and could benefit from further exploration.

**Essential References Not Discussed:**

Some related works are not described, like the papers include critique ability concepts
1. A Survey on LLM-as-a-Judge
2. From Generation to Judgment: Opportunities and Challenges of LLM-as-a-judge
3. CriticEval: Evaluating Large Language Models as Critic

**Experimental Designs Or Analyses:**

The experimental design is comprehensive and addresses the key questions about the effectiveness of the proposed method. The use of multiple synthetic evaluators (e.g., GPT-4, Skywork-8B) and fine-tuning experiments adds robustness to the findings. The alignment between theoretical savings and empirical results is a strong point. However, the experiments primarily focus on head-to-head comparisons, and it would be valuable to test the method on more diverse evaluation tasks. Additionally, some results (e.g., the saving ratios in Table 1) could be better contextualized to highlight their practical implications.

**Methods And Evaluation Criteria:**

The proposed method and evaluation criteria are well-suited to the problem at hand. The use of Chatbot Arena and MT Bench as benchmarks ensures the relevance and applicability of the results. The incorporation of synthetic feedback and the focus on reducing human annotation costs align with the goals of scalable and efficient LLM evaluation. However, the paper could expand on how the method might generalize to other evaluation setups, such as multi-model ranking or fine-grained assessments.

**Other Comments Or Suggestions:**

Typos: In Section 5.5, the phrase "introduce more significant savings" could be rephrased for clarity.

**Other Strengths And Weaknesses:**

Strengths:

The method is original and addresses a critical bottleneck in LLM evaluation—reducing human annotation costs.
The theoretical framework is solid, and the empirical results strongly support the claims.
The introduction of the human annotation saving ratio as a predictive metric is a useful and practical contribution.
Weaknesses:

The focus is narrow, primarily on head-to-head comparisons, limiting the generalizability of the results.
Some theoretical details, while likely correct, could be presented more clearly.
The paper assumes access to high-quality synthetic feedback, which may not always be feasible in practice.

**Questions For Authors:**

No questions

**Relation To Broader Scientific Literature:**

The paper is well-situated within the broader literature on LLM evaluation. It builds on prior work on synthetic feedback (e.g., LLM-as-a-judge) and variance reduction techniques (e.g., control variates in Monte Carlo sampling).
Moreover, this paper also connect to some famous concepts like critique ability and reward models.
The connections to recent benchmarks like Chatbot Arena and MT Bench are appropriate and timely. However, the paper could benefit from a deeper discussion of related methods for reducing human annotation costs, such as active learning or adaptive sampling, to highlight its unique contributions.

**Theoretical Claims:**

The theoretical claims, particularly those concerning variance reduction using control variates, appear sound. The proofs provided in Section 4.1 are logically structured, and the derivations seem correct at a high level. However, I did not verify all mathematical details rigorously, and some minor steps in the derivations (e.g., bias analysis in Equation 2) could benefit from additional clarification. While the claims are likely correct, their presentation could be more transparent for broader accessibility.

---

> ### Author Rebuttal · Authors · 2025-04-01
>
> Thank you for your positive feedback. We address your comments below.
> > **Weakness 1: Beyond head-to-head comparisons**
>
> Our theory directly applies to many other evaluation tasks, such as single response evaluation, where a human gives scores to a single LLM generation, instead of giving preference to two LLM generations. However, there are limited public datasets available for these tasks, and collecting such a dataset might require a significant amount of human effort which might be beyond the scope of our paper. Nonetheless, we believe that this will be an interesting future effort.
>
> That said, we conduct an additional experiment in the single response evaluation setting. We utilize the validation split of the HelpSteer2 dataset as our benchmark. This split consists of 1.04K samples, each containing a prompt, a response, and five human-annotated attributes: helpfulness, correctness, coherence, complexity, and verbosity. Each attribute is scored from 0 to 4, with higher scores indicating better performance. Our focus is on the helpfulness attribute, as it is the primary metric that reward models are typically trained to evaluate. We employ the Control Variates Evaluation method to predict the average helpfulness score.
>
> The human annotation saving ratio is shown in the table below:
>
> |     Model                         | GRM-2B | Skywork-8B | ArmoRM-8B | GPT-4o |
> |------------------------------|--------|------------|-----------|--------|
> | Saving | 10.3%  | 21.0%      | 14.1%     | 27.4%  |
>
> The result above indicates the perspective of Control Variates Evaluation in single-response evaluation. To our best knowledge, this is the only public dataset with high-quality human annotation for single-response evaluation. We will include this experiment in the camera-ready version.
>
> > **Weakness 2: Theoretical details, bias analysis in Equation (2)**
>
> This is discussed in https://artowen.su.domains/mc/Ch-var-basic.pdf, Page 32. We will expand the clarification in Appendix in the final version for completeness.
>
> > **Weakness 3: Assume access to high-quality synthetic feedback**
>
> Our method is effective as long as there is a non-zero correlation between human and synthetic evaluations. While high-quality synthetic feedback is ideal due to its typically strong correlation, evaluations from a small reward model—though highly biased regarding human preferences—can still yield satisfactory performance in Control Variates Evaluation, as shown in Table 1. Ultimately, we believe the correlation requirement will diminish as AI systems continue to progress.
>
> > **Relation To Broader Scientific Literature: related methods for reducing human annotation costs**
>
> To our best knowledge, Control Variates Evaluation is the first unbiased LLM evaluation method with variance reduction. There are indeed other methods to reduce human annotations, such as Active Evaluation Acquisition (AEA) [1]. However, AEA might introduce bias to the evaluation because choosing a subset of human annotation data causes distribution shift in the evaluation dataset. In addition, AEA requires training of the neural process, while finetuning is optional in our method.
>
> Furthermore, we can combine active learning and control variates evaluation to further reduce human annotations in LLM evaluation. To be specific, we can first apply active learning to select a representative subset of prompts for evaluation, and then run control variates evaluation on this subset. The downsides of this combination are:
> 1) The evaluation will be biased because strategic sampling of responses causes distribution shift with respect to the original evaluation dataset.
> 2) Active learning–based approaches like AEA [1] require an additional training procedure, which relies on existing human annotations.
>
> We will add this discussion in the final version of our paper.
>
> [1] Li, Yang, et al. "Active Evaluation Acquisition for Efficient LLM Benchmarking." arXiv preprint arXiv:2410.05952 (2024).
>
>
> > **Essential References Not Discussed**
>
> We will include these papers in the camera-ready version.
>
> > **Typos**
>
> We will change it to "improve the human annotation saving ratio."

---

### Official Review · Reviewer_yoN8 · 2025-03-10

**Overall Recommendation:** 4

**Summary:**

-Paper proposes Control variates evaluation --- the goal being to reduce the cost of LLM evaluations

-It does so using a principled statistical approach that combines human annotations with synthetic feedback (i.e. LLM as a judge).

-Specifically, the synthetic feedback is the control variate to reduce the variance of limited human evals.

-Paper shows the generic & fine-tuned approach reduces the number of human annotations.

---
Update after rebuttal

Thank you for the detailed responses.

Adding these to the revised paper would strengthen the paper and ensure clarity.

That said, I retain my score and my positive assessment of the paper - best of luck :)

**Claims And Evidence:**

The claims made in the paper are generally well-supported by theoretical analysis and experiments.

- Good theory in Sec 4 and formal proofs in the appendices.

- The unbiasedness is shown empirically and theoretically

- The main claim of human annotation savings is shown across a variety of settings.

------------

- Only doubt is whether the claims are generalizable beyond the head-head settings.

**Essential References Not Discussed:**

It would be useful for the paper to position a bit better against alternative variance reduction approaches. For instance, one could use active learning to decide which limited set needs human eval and which can remain as synthetic.

**Experimental Designs Or Analyses:**

As mentioned above experimental designs and analyses are sound:

- Well structured, uses multiple synthetic evaluators, datasets and fine-tuning vs no fine-tuning

- As mentioned before, and acknowledged by the authors it’s unclear how this generalizes to other eval setups.

**Methods And Evaluation Criteria:**

-Use of control variates is well motivated from the perspective of variance reduction

-Good use of established benchmarks chatbot arena and MT-bench

-Nice evaluation across different sizes of models

-Q: Asked later --- are their other variance reduction approaches that could be baselined against. i.e. is control variates the best?

**Other Comments Or Suggestions:**

-It would be useful to add some computational costs, e.g. cost of fine-tuning vs human annotation costs incurred

-Maybe adding an appendix fleshing out how the method could be extended beyond head-head comparison

**Other Strengths And Weaknesses:**

Strengths:

-	Tackles an important problem with a general approach --- variety of evaluators

-	Convincing empirical results for an important problem

-	Nice theoretical links

Weaknesses:

-	Only head-to-head evaluation tasks considered

-	No comparison with other variance reduction methods

-	Assessment of computational costs

**Questions For Authors:**

-Is there a minimum number of human annotations needed for the control variate to be reliable?

-Figs 7 and 8 show variance for the annotation saving ratio across different LLM pairs? Do you have insights into the characteristics of different LLM pairs and why certain ones are more amenable and have higher savings?

-Is it possible that alternative variance reduction approaches like active learning could fit into this paradigm and how would they differ from the proposed approach

**Relation To Broader Scientific Literature:**

The paper situates itself well within several research areas: LLM as a judge, efficiency LLM evals and control variates.

**Theoretical Claims:**

-I’m not a theory expert, but the proofs seem correct

-One thing I noted is the assumption of the strong correlation between human and synthetic annotations being important. Perhaps some ablations or analysis where the relationship is weak would be useful

---

> ### Author Rebuttal · Authors · 2025-04-01
>
> Thank you for your positive feedback. We address your        questions and comments below.
> > **Weakness 1 & Comment 2: Only head-to-head evaluation tasks**
>
> Public datasets for other evaluation tasks are limited, and collecting such data may require significant human effort, which is beyond our paper's scope. However, we believe that this will be an interesting future effort and currently our theory directly applies to many other tasks, such as single response evaluation, where human scores are given to individual LLM outputs rather than comparing two responses.
>
> Therefore, we conduct an experiment in this setting using validation split of HelpSteer2 as our benchmark.
> The human annotation saving ratio is shown in the table below:
>
> |     Model                         | GRM-2B | Skywork-8B | ArmoRM-8B | GPT-4o |
> |------------------------------|--------|------------|-----------|--------|
> | Saving | 10.3%  | 21.0%      | 14.1%     | 27.4%  |
>
> To our best knowledge, this is the only public dataset with high-quality human annotation for single-response evaluation. We will include this experiment in the camera-ready version.
>
> > **Weakness 2 & Question 3: No comparison with other variance reduction methods, e.g. active learning**
>
> To our best knowledge, Control Variates Evaluation is the first unbiased LLM evaluation method with variance reduction. There are indeed other methods to reduce human annotations, such as Active Evaluation Acquisition (AEA) [1]. However, AEA might introduce bias to the evaluation because choosing a subset of human annotation data causes distribution shift in the evaluation dataset. In addition, AEA requires training of the neural process, while finetuning is optional in our method.
>
> We can also combine active learning and control variates evaluation to further reduce human annotations in LLM evaluation. To be specific, we can first apply active learning to select a representative subset of prompts for evaluation, and then run control variates evaluation on this subset. The downsides of this combination are:
> 1) The evaluation will be biased because strategic sampling of responses causes distribution shift with respect to the original evaluation dataset.
> 2) Active learning–based approaches like AEA [1] require an additional training procedure, which relies on existing human annotations.
>
> [1] Li, Yang, et al. "Active Evaluation Acquisition for Efficient LLM Benchmarking." arXiv preprint arXiv:2410.05952 (2024).
>
> > **Weakness 3 & Comment 1: Assessment of computational costs**
>
> In our experiment, fine-tuning reward models, such as the 7B model, can be performed locally using four H100 GPUs with 80GB of GRAM. The evaluation cost for GPT-4o is approximately $0.0035 per annotation. While the exact cost of human annotation is unknown, we believe it is significantly more expensive, by orders of magnitude. We will include this discussion in the final version.
>
> > **Question 1: Minimum number of human annotations needed for the control variate to be reliable**
>
> Theoretically, the reliability of Control Variates Evaluation is independent of number of human evaluation since it is unbiased. That said, if we want to compare with purely synthetic evaluation in practice, then the minimum number of human annotations required depends on the point at which the variance of Control Variates Evaluation is lower than the square of the synthetic evaluation bias. Since this threshold is influenced by the intrinsic variance of human annotations on a given evaluation dataset, it must be determined empirically. However, as shown in Figures 4 and 5, Control Variates Evaluation with just 200 human annotations already achieves significantly lower error than synthetic evaluation across all experiments. This is a relatively small number compared to the scale of popular LLM benchmarks such as MT Bench and Chatbot Arena.
>
> > **Question 2: Interpretation of Figure 7 and 8**
>
> They show the human annotation saving ratio (please refer to Page 4, right column of Line 180) of different LLM pairs. Factors influencing this ratio include the architecture of the response generator LLMs, the datasets used for training the generators, and whether an LLM is distilled from another LLM. Exploring these factors in detail is a focus for our future research.
>
> > **Theoretical Claims: Assumption of the strong correlation between human and synthetic annotations**
>
> We would like to clarify that our method works as long as the correlation is non-zero. While a strong correlation is ideal, a weak correlation can still lead to significant human sample savings: we observe from experiments (Table 1) that weak reward models can still achieve satisfactory correlation and thus good human annotation saving in Control Variates Evaluation.
>
> > **Supplementary Material: More details on prompt templates**
>
> We will attach the specific prompt template for completeness in the final version.

---

> > ### Comment · Reviewer_yoN8 · 2025-04-01
> >
> > Thank you for the detailed responses.
> >
> > Adding these to the revised paper would strengthen the paper and ensure clarity.
> >
> > That said, I retain my score and my positive assessment of the paper - best of luck :)

---

> > > ### Author Response · Authors · 2025-04-04
> > >
> > > We are glad to know that our rebuttal improved the technical content and clarity of the paper, and we will make sure to incorporate them into the final version. Thank you again for your positive assessment of our paper!

---

### Official Review · Reviewer_1bee · 2025-03-18

**Overall Recommendation:** 4

**Summary:**

The paper proposes Control Variates Evaluation, a method to reduce human annotation costs in evaluating large language models (LLMs) while maintaining unbiased results. By combining synthetic feedback from LLMs, the method achieves variance reduction in win rate estimation. The approach is theoretically grounded in the control variates technique and has been validated on benchmarks like Chatbot Arena and MT Bench, demonstrating human annotation savings. This general and scalable method offers a cost-effective alternative to full human evaluation without compromising reliability.

**Claims And Evidence:**

The claims made in the paper are well-supported by clear and convincing evidence:

- The paper shows the method is unbiased and achieves variance reduction. Experimental results demonstrate that Control Variates Evaluation matches human evaluation accuracy with fewer annotations, and the mean square error converges to near zero, indicating negligible bias.

- The paper establishs the properties of the control variates method and its application to LLM evaluation.

- The paper presents human annotation saving ratios across different evaluators and benchmarks, showing consistent savings. The experimental results validate that the theoretical saving ratios match practical variance reduction, with figures demonstrating the effectiveness across various evaluators and benchmarks.

**Essential References Not Discussed:**

Given the generality of Control Variates, I believe the paper is self-explanatory and does not require further elaboration from other sources. However, it might be beneficial to cite and discuss additional papers that utilize Control Variates for different purposes, such as [R1].

[R1] Training Chain-of-Thought via Latent-Variable Inference

**Experimental Designs Or Analyses:**

I have reviewed the experimental sections. The overall experimental designs are well-structured and comprehensive. The experiments are well-aligned with the theoretical results.

**Methods And Evaluation Criteria:**

The method is theoretically grounded in the control variates technique from statistics, which is a well-established variance reduction method in Monte Carlo sampling. Thus combining human annotations with synthetic feedback in a principled way makes sense for reducing annotation costs while preserving unbiasedness

**Other Comments Or Suggestions:**

Typos:

- Line 186: It should be $u_{\hat{z}}$ instead of $\hat{u}_{z}$.

- Line 601: It should be $\frac{1}{n}$ instead of $\frac{1}{n}(1-\rho^{2})$

**Other Strengths And Weaknesses:**

Strengths:
- The paper presents a novel application of the control variates technique from statistics to the problem of LLM evaluation. This creative combination of an established statistical method with modern LLM evaluation challenges addresses a gap in the field. The introduction of the human annotation saving ratio as a metric provides a clear way to quantify the method's effectiveness.

- The extensive experimental validation across multiple evaluators and benchmarks effectively demonstrates the robustness and generalizability of the method. The inclusion of both pretrained and fine-tuned synthetic evaluators further enhances the practical relevance of the findings. Particularly, I was surprised to see in Figure 6 how closely the shifted Control Variates aligned with human error.

- The paper is well-structured and clearly written. The figures and tables effectively illustrate the method and results, and the appendix provides additional details for reproducibility.

Weaknesses:

- The method and the experiments focus primarily on head-to-head win rate estimation. The paper does not explore other evaluation metrics or more complex scenarios like multi-model ranking.

**Questions For Authors:**

- Regarding the fine-tuned section, could the authors elaborate on how they ensured that the responses in the evaluation dataset are out-of-distribution with respect to the fine-tune dataset? This is crucial because if the datasets share a similar distribution, the comparison would no longer be fair.
- Can authors discuss how to generalize Control Variates to other evaluations beyond head-to-head win rates.

**Relation To Broader Scientific Literature:**

I believe Control Variates can serve as a valuable technique for LLM evaluation in the future. Although Control Variates is not a new concept, to my knowledge, this is the first time it has been introduced into the field of synthetic LLM evaluation.

**Theoretical Claims:**

Yes, I have thoroughly reviewed the theoretical proofs. The general proofs are correct, as they directly follow from the principles of Control Variates. However, I have one question:

- The general application of Control Variates does not require the sampled $x$ to follow a uniform distribution. Unless, in your design, the uniform distribution is an inherent assumption in the evaluation objective (win rate)—specifically, to estimate the global win rate equally weighted across all prompts, rather than reflecting preferences for a specific distribution. I would appreciate it if the authors could clarify this point.

---

> ### Author Rebuttal · Authors · 2025-04-01
>
> Thank you for your careful review and constructive feedback. We address your comments and questions below.
> > **Theoretical Claim: The general application of Control Variates does not require the sampled $x$ to follow a uniform distribution.**
>
> Our method can be applied directly to the setting where $X$ follows a non-uniform distribution. We assume the prompt $x$ to be sampled uniformly from $X$ because that is what people usually do when evaluating LLMs in practice. We will add a clarification in the camera-ready version.
>
> > **Essential References Not Discussed: [R1] Training Chain-of-Thought via Latent-Variable Inference**
>
> We will include this paper in Section 2.3 in the camera-ready version.
>
> > **Weaknesses: Focus primarily on head-to-head win rate estimation**
>
> There are currently no public datasets available for testing multi-model ranking, and collecting such a dataset might require a significant amount of human effort which might be beyond the scope of our paper. However, we believe that this will be an interesting future effort and currently our theory directly applies to the multi-model ranking setting.
>
> That said, we conduct an additional experiment in the single response evaluation setting, where a human gives scores to a single LLM generation, instead of giving preference to two LLM generations.
>
> We utilize the validation split of the HelpSteer2 dataset as our benchmark. This split consists of 1.04K samples, each containing a prompt, a response, and five human-annotated attributes: helpfulness, correctness, coherence, complexity, and verbosity. Each attribute is scored from 0 to 4, with higher scores indicating better performance. Our focus is on the helpfulness attribute, as it is the primary metric that reward models are typically trained to evaluate. We employ the Control Variates Evaluation method to predict the average helpfulness score.
>
> The human annotation saving ratio is shown in the table below:
>
> |     Model                         | GRM-2B | Skywork-8B | ArmoRM-8B | GPT-4o |
> |------------------------------|--------|------------|-----------|--------|
> | Saving | 10.3%  | 21.0%      | 14.1%     | 27.4%  |
>
> The result above indicates the perspective of Control Variates Evaluation in single-response evaluation. To our best knowledge, this is the only public dataset with high-quality human annotation for single-response evaluation. We will include this experiment in the camera-ready version.
>
> > **Typos**
>
> Thanks for pointing them out. We will fix them in the final version.
>
> > **Question 1: How to ensure that the responses in the evaluation dataset are out-of-distribution with respect to the fine-tune dataset?**
>
>
> As we discussed in Section 5.1, the evaluation dataset is out-of-distribution in the sense that the evaluated model's response is excluded from the training dataset for fine-tuning. In the final version, we will reference Section 5.1 in the paragraph "(Optional) Synthetic Evaluator Fine-Tuning" on Page 5 for added clarity.
>
> > **Question 2: Generalize Control Variates to other evaluations beyond head-to-head win rates**
>
> This is addressed in the "Weaknesses" section above.

---

> > ### Comment · Reviewer_1bee · 2025-04-04
> >
> > I have carefully reviewed the authors' responses as well as the other reviewers' comments. Most of my concerns have been addressed, so I am raising my score from 3 to 4 and now lean toward acceptance.

---

> > > ### Author Response · Authors · 2025-04-04
> > >
> > > Thank you for raising your score! We appreciate your feedback, and we are happy to know that our rebuttal addressed your concerns. We will reflect the changes in the final version of our paper.

---

### Decision · Program_Chairs · 2025-05-01

**Decision:**

Accept (poster)

**Comment:**

The paper proposes Control Variates Evaluation, a method to reduce human annotation costs in evaluating large language models (LLMs) while maintaining unbiased results. The approach leverages synthetic feedback from LLMs, combined with human annotations, to reduce annotation costs while maintaining evaluation reliability. The method is empirically validated against benchmarks such as Chatbot Arena and MT Bench demonstrating that Control Variates Evaluation matches human evaluation accuracy with fewer annotations, and the mean square error converges to near zero, indicating negligible bias.

Strengths:
Strong results: The paper presents human annotation saving ratios across different evaluators and benchmarks, showing consistent savings. The experimental results validate that the theoretical saving ratios match practical variance reduction, with figures demonstrating the effectiveness across various evaluators and benchmarks.
The method is theoretically grounded in the control variates technique from statistics, which is a well-established variance reduction method in Monte Carlo sampling. Thus combining human annotations with synthetic feedback in a principled way makes sense for reducing annotation costs while preserving unbiasedness.
The method is original and addresses a critical bottleneck in LLM evaluation—reducing human annotation costs.

Can you please discuss how much this methods differs to PPI++?
Anastasios N. Angelopoulos, John C Duchi, and Tijana Zrnic. PPI++: Efficient PredictionPowered Inference. arXiv preprint arXiv:2311.01453, 2023